# Study of the Interface between Wild Bird Populations and Poultry and Their Potential Role in the Spread of Avian Influenza

**DOI:** 10.3390/microorganisms11102601

**Published:** 2023-10-21

**Authors:** Luca Martelli, Diletta Fornasiero, Francesco Scarton, Arianna Spada, Francesca Scolamacchia, Grazia Manca, Paolo Mulatti

**Affiliations:** 1Istituto Zooprofilattico Sperimentale delle Venezie, 35020 Legnaro, Italy; lmartelli@izsvenezie.it (L.M.); fscolamacchia@izsvenezie.it (F.S.); gmanca@izsvenezie.it (G.M.); 2SELC Soc. Coop., 30175 Venice, Italy; scarton@selc.it (F.S.); arianna.spada@gmail.com (A.S.)

**Keywords:** HPAI, wild birds, ornithocenosis, bridge hosts, spillover, wild–domestic interface, camera trap, ornithological transects

## Abstract

Water birds play a crucial role in disseminating and amplifying avian influenza viruses (AIVs) in the environment. However, they may have limited interactions with domestic facilities, raising the hypothesis that other wild birds may play the bridging role in introducing AIVs into poultry. An ornithocoenosis study, based on census-transect and camera-trapping methods, was conducted in 2019 in ten poultry premises in northeast Italy to characterize the bird communities and envisage the species that might act as bridge hosts for AIVs. The data collected were explored through a series of multivariate analyses (correspondence analysis and non-metric multidimensional scaling), and biodiversity indices (observed and estimated richness, Shannon entropy and Pielou’s evenness). The analyses revealed a high level of complexity in the ornithic population, with 147 censused species, and significant qualitative and quantitative differences in wild bird species composition, both in space and in time. Among these, only a few were observed in close proximity to the farm premises (i.e., Magpies, Blackbirds, Cattle Egrets, Pheasants, Eurasian Collared Doves, and Wood Pigeons), thus suggesting their potential role in spilling over AIVs to poultry; contrarily, waterfowls appeared to be scarcely inclined to close visits, especially during autumn and winter seasons. These findings stress the importance of ongoing research on the wild–domestic bird interface, advocating for a wider range of species to be considered in AIVs surveillance and prevention programs.

## 1. Introduction

Northeastern Italy is known for its high concentration of poultry farms and the presence of extensive and widespread wetlands in close proximity to the intensive poultry areas, which serve as ecological niches for various Orders of wild birds. Over the years, due to these key characteristics, this area has been recurrently affected by several severe avian influenza (AI) epidemics, with devastating impacts on the poultry sector [1,2,3]. The most recent Italian epidemics occurred during 2021–2022 (which involved 317 domestic farms and caused more than 13 million culled birds) [4] and during 2022–2023 (marked by the deaths of hundreds of wild birds and 40 outbreaks in the domestic sector) [5], proving that AI remains a serious problem that needs further countermeasures.

The repeated occurrences of the AI outbreaks prompt inquiries regarding the mechanisms behind the emergence of these viruses. The questions encompass the virus origin, the pathways through which it enters Europe and its introduction into poultry farms. In fact, until 2003, most transmission events from wild to domestic birds involved low pathogenicity avian influenza (LPAI) viruses, some of which later mutated into high pathogenicity avian influenza (HPAI) viruses within the domestic populations. However, since 2005, the spread of HPAI H5Nx viruses of the Goose/Guangdong lineage from China to other areas in Asia, the Middle East and subsequently to Europe, Africa, and North America, has occurred in a series of intercontinental waves facilitated by the movement of migratory bird populations [6,7,8].

Avian influenza viruses (AIVs) have been isolated from more than 90 wild bird species representing 13 different Orders [9,10]. Free-living water birds belonging to the Orders of Anseriformes and Charadriiformes are most commonly associated with AIVs and they are generally considered as the maintenance hosts in their natural ecology [11,12]. On the other hand, terrestrial wild bird species that have shown natural susceptibility to infection do not appear to be a primary source of the virus or a reservoir for the disease [12,13,14]. However, over time it has been recognized that some terrestrial bird species may play an important role in facilitating the local transmission of AIVs, including high pathogenic strains, to domestic poultry [11,15,16,17]. The European Food Safety Authority (EFSA) itself compiled a list of fifty wild bird species to consider as targets for passive surveillance in the European Union. The inclusion criterion adopted by EFSA considers the species with an HPAI H5N1 prevalence ≥ 0.4% during the period of 2005–2017 [18]. The final list includes, in addition to species of Charadriiformes and Anseriformes, birds belonging to the other seven Orders, including Passeriformes, Accipitriformes, Falconiformes, and Strigiformes.

Several studies on migratory bird movements have shown that birds belonging to the Charadriiformes and Anseriformes Orders are involved in the wide-scale spread of the virus [19,20,21]. Further supporting this hypothesis, phylogenetic analyses of both LPAI [22] and HPAI viruses [6,23,24] have identified intercontinental movements of viral strains through the movements of migratory waterfowl populations. However, it is noted that these wild birds rarely get close to the domestic poultry premises, especially in the case of large-scale farms, mainly due to their ecological needs. Indeed, as they are not granivores, these species normally seek extensive wetland areas for their foraging and resting activities [25].

It emerges that, to fully understand the eco-epidemiology of the AI disease, a link that justifies the recurrent and widespread occurrence of spillovers into the domestic poultry sector is still missing and needs to be elucidated [11,15]. In this scenario, terrestrial wild birds that share habitats with waterfowls, and that also visit poultry farms, have been proposed as the potential actors in the viral transmission between the maintenance hosts and the poultry sector [26]. These species can be defined as ‘bridge hosts’, that is, species that are unable to maintain the infection in nature but are sufficiently competent to successfully transmit the pathogen from the reservoir populations to the target animal species [11]. Specifically, AIVs can be transmitted by bridge hosts through either direct and indirect contact, mechanical spread, or contamination of resources (such as feed, litter, or other consumables). The hypothesis emerges that a bridge host might compensate for the scarcity of productive encounters (in terms of effective transmission of infection) between the reservoir populations and the target species, increasing the chances of potentially effective contacts, operating on the spatial, temporal, and behavioral dimensions [27]. For AI, this mechanism typically operates on the spatial dimension, which means that these species establish a spatial connection between distinct areas where the maintenance and target host populations are found, acting as ‘spatial vectors’ of the disease [11,27]. Although the role of the bridge host is fairly intuitive, understanding which species, and to what extent, actually fulfil this role can be quite challenging, considering the wide range of farming realities they can interface with. Gaining new insights on the potential AI bridge host(s) would be extremely advantageous for adjusting the risk assessments of the virus introduction and spread into the domestic populations, thus allowing for the application of effective countermeasures.

Herein, we present an ornithocenosis study to characterize the wild bird communities living nearby of some poultry farms located in northeastern Italy. The study was developed through two parallel activities: (i) the installation of camera traps within the farm premises, and (ii) the census conducted using transects to identify and count wild birds in the farms’ surrounding environment. The data collected through these activities were then used to perform: (i) some descriptive analyses to quantitatively and qualitatively define the wild bird populations, (ii) an assessment of the biodiversity level of the study areas, and (iii) an evaluation of the possible spatial and temporal variations in the composition of the wild bird communities. The ultimate goal is generating new hypotheses on the potential species that could act as a bridge between the wild water bird and the domestic poultry populations, allowing for a deeper understanding of the eco-epidemiology of the disease.

## 2. Materials and Methods

### 2.1. Definition of the Study Area

The on-field activities were coordinated by the Veterinary Epidemiology Laboratory of the Istituto Zooprofilattico delle Venezie in collaboration with the SELC Cooperative Society located in Marghera, Venezia.

The study involved 10 poultry farms, characterized by different reared species and production types, located in northeastern Italy. The activities were conducted from January to December 2019. The farms were selected within the geographic areas characterized by a very high density of domestic poultry farms, or in proximity to important wetland areas, to cover a variety of environments to allow for the identification of potential different spatiotemporal patterns of the wild–domestic interface. The study area matches the one most affected during the 2016–2018 epidemic sustained by a HPAI H5N8 virus (Figure 1).

A brief description of the farms included in the study is hereafter reported.

Five fattening turkey farms were included in the study: FE-01, RO-01, and VE-02 located close to the Adriatic coast, and VR-01 and VR-02, which are situated further inland, in the province of Verona. FE-01 is a large farm with a potential capacity of 94,000 meat turkeys and is surrounded by extensive arable land. An irrigation canal is present nearby, and an artificial reservoir is located approximately 1.6 km away. RO-01 has a potential capacity of 48,000 animals. It is surrounded by areas predominantly dedicated to arable crops. About 1 km away, there is an artificial canal that flows through the Polesine area in the province of Rovigo, and a few valleys used for fishing activities in the Po river delta. VE-02 is located less than 600 m from the Venice lagoon, in proximity to a fishing valley and an industrial poplar plantation. The surrounding cultivated lands are enclosed by hedges. Its potential capacity is 36,000 animals. VR-01 is located at a short distance from the course of the Adige River, within an agricultural setting with scattered residential areas, and a riparian woodland nearby. This is the smallest farm among those selected, with a potential capacity of 15,000 animals. VR-02 is situated in areas dominated by arable lands and orchards, and it is located less than 2 km from the A4 Venice–Milan highway. It is a medium-large intensive farm with a potential capacity of 40,000 animals.

Four laying hen farms, named BO-01, FE-02, MN-02, and VE-01, were included in the study. BO-01 is an organic farm presenting two outdoor paddocks. The surrounding areas are mostly dedicated to arable crops, and some uncultivated and forage areas. It is located not far from a small settlement and an irrigation canal. Its total capacity is 129,000 laying hens. FE-02 is located in areas dedicated to arable farming, bordered by two large artificial canals. This is the largest site included in the study and it can accommodate up to 1 million animals. The areas surrounding the MN-02 farm are mostly dedicated to arable farming, while a small, wooded area marks its boundary. It can accommodate a maximum of 400,000 laying hens. VE-01 is located in the countryside near a populated center. The Musone Vecchio river is situated just over 1 km north of the farm, and the surrounding area is characterized by agricultural land with the presence of trees and an artificial poplar plantation. It is the smallest among the selected layer farms, with a capacity of 68,000 animals.

One fattening duck farm was included in the study: MN-01. It is a farm with a potential capacity of 30,000 ducks. It is located in an area primarily dedicated to arable farming, and approximately 1 km northwest, a large quarry for stone materials is present, while some urbanized areas are present in the south.

### 2.2. Ornithological Transects

Throughout the entire year of 2019, each selected farm was visited twice every month to conduct a census using transects. Transects are commonly used in ornithological studies to survey bird populations and determine the structure of bird communities in defined areas [28,29]. They involve walking along pre-established routes traced on a geographical map. Three transects of approximately 500 m each (i.e., 1500 m in total per farm) were defined within a buffer of approximately 1 km radius from the perimeter of the farm (Appendix A). The transects were positioned to representatively delineate the land use in the vicinity of the farms. Although the predominant habitat consisted mainly of intensive agricultural land, the transects were designed to cover other types of landscapes as well, including watercourses, woods, and semi-urban areas.

The bird census was conducted through visual and auditory observations. Only observations made within 100 m of the transects were considered valid for the study. Therefore, each ornithological transect included observations in a total area of approximately 10 hectares (500 × 100 m per side). In general, this surface area is considered sufficiently large to detect a number of birds that adequately represents the bird community around that area [29].

### 2.3. Camera-Trapping Activities

The sampling protocol involved the use of two camera traps (Browning^®^ Dark OPS HD 940) per farm oriented to cover two separate sectors within the internal area of the farms’ perimeter, where the presence of wild birds was considered more likely. The position of each camera trap from the premises was within a distance of 20 m (Appendix A). A total of 20 camera traps were set up to operate simultaneously, remaining active 24 h per day, during the year of 2019. The camera traps were equipped with a passive infrared sensor capable of detecting both movements and changes in temperature in the surveyed area [30]. The images were recorded on a 32-gigabyte Secure Digital card, and they were downloaded and subsequently deleted approximately every 4–5 weeks. The photos depicting people were deleted within 24 h, following the current Italian privacy regulations [31]. Whenever the identification of the species by the ornithologists was not entirely certain, such as when the birds’ identification was only possible at the genus level due to unsuitable environmental conditions, those records were classified as ‘uncertain’, and not included in subsequent analyses.

The data collection activity through camera-trapping surveys was conducted to observe and assess the species and frequency of wild birds visiting the farms. Therefore, no baits (such as food or olfactory attractants) were employed to avoid potential biases in the interpretation of the results. Additionally, the daily presence/absence of species near each farm were considered, aggregating the events recorded from both cameras per farm, to avoid overestimations due to repeated captures of the same individuals by a single camera trap or neighboring camera traps.

### 2.4. Data Analysis

#### 2.4.1. Descriptive Analysis of Transect Data

The data collected through the ornithological transects were explored to qualitatively and quantitatively describe the bird populations observed within the study area. The total daily abundances of wild birds were considered to assess possible significant differences in the recorded values between the farming sites and across the months of the year. The various species were then ranked based on their relative abundance to the total observations recorded by the ornithologists along the transects (Appendix A).

#### 2.4.2. Multivariate Analysis

Correspondence analysis (CA) is a multivariate analysis that reveals the relationships between the categories within two groups of variables [32]. The CA was applied to investigate the potential associations between (i) species and observation sites, and (ii) between species and months of observation. To assess whether a significant relationship exists between these categorical variables (species vs. sites, and species vs. months), the chi-square test was applied [32]. Only species observed with a percentage greater than 1% of the total identified birds were included in this analysis.

For the classifying and ordering of observation sites based on species richness and abundance detected along the transects, non-metric multidimensional scaling (nMDS) and cluster analysis techniques were used, applying the Bray–Curtis index. The nMDS allows for a graphical representation of the dissimilarities between the farms with respect to species and their abundances. To evaluate the goodness of the analysis, the stress index was assessed [32,33,34]. The dendrogram produced by applying a cluster analysis simplifies the interpretation of the calculated diversity between sites, according to the Bray–Curtis indices, allowing the visualization of farms that cluster together based on the qualitative–quantitative similarity of the bird populations present [32].

#### 2.4.3. Biodiversity Measures

To characterize the structure of the bird populations living in the surrounding environments of the study sites, some α-diversity indices were calculated and summarized in Table 1. α-diversity refers to the level of complexity of a specific community of organisms present in an area of interest [35,36]. The indices were calculated on the data obtained through the ornithological transects and grouped by the clusters identified through the multi-dimensional scaling methods presented above, to characterize and compare the identified wild bird communities. To obtain reliable and comparable information on the diversity levels, a series of indices that account for different aspects of the studied communities were considered, to overcome partial and imprecise conclusions that might arise from the evaluations based on just individual biodiversity indices [36,37].

#### 2.4.4. Descriptive Analysis of Camera-Trapping Data

A descriptive analysis and a CA of the data collected through camera trapping were performed to define the number of farms in which each species was photographed and to highlight any potential relationship between the visiting wild species and poultry farms. To quantify the tendency of each species to approach the farming sites and evaluate potential differences between the clusters identified by the multivariate analysis, the number of days in which each species was camera-trapped was considered for each cluster and each month of the year. This was performed taking into account the target species list for passive surveillance of avian influenza compiled by EFSA (Appendix A). This list indicates which wild bird species in the European Union are more likely to be exposed to the HPAI H5 virus and to suffer lethal infection [18].

#### 2.4.5. Statistical Tests and Software

Significant differences between the distributions of abundances and biodiversity indices were assessed through the non-parametric Kruskal–Wallis test [41], considering a significance level of 5%. In the presence of statistically significant differences, the post hoc unpaired Wilcoxon test with Benjamini and Hochberg correction for multiple testing [42] was applied to identify pairs of distributions with statistically significant differences. 

All data cleaning and manipulation, descriptive statistical analyses, and graphics were performed using the R software 4.2.2 [43] and RStudio [44], along with the following packages: DiversitySeq [45] for α-diversity indices, vegan [46] for cluster analysis and nMDS, ca [47] for the correspondence analysis, factoextra [48] and ggplot2 [49] for the graphical plots, stats [43] for the statistical tests.

## 3. Results

### 3.1. Ornithological Transects

Between January and December 2019, a total of 52,794 birds belonging to 146 species and 19 different Orders were observed in the 10 farms (Appendix A). Overall, the range of the number of wild birds observed daily along the ornithological transects varied from a minimum of n = 20 (on 8 March 2019 at the VR-01 farm) to a maximum of n = 2709 individuals (on 22 October 2019 at the RO-01 farm). Significant differences were observed among the abundance distributions calculated per farm (H_9_ = 53,940, *p*-value < 0.0001), while the differences observed in the monthly distributions were not significant (H_11_ = 3.109, *p*-value = 0.989; Figure 2). The farms with the highest abundance of observed birds (median, m; interquartile range, IQR) were FE-01 (m_FE-01_ = 364, IQR_FE-01_ = 364.25) and BO-01 (m_BO-01_ = 289.50, IQR_BO-01_ = 183), while VE-01 (m_VE-01_ = 80.50, IQR_VE-01_ = 72.75), VE-02 (m_VE-02_ = 106, IQR_VE-02_ = 317), and VR-02 (m_VR-02_ = 108.50, IQR_VR-02_ = 108.50) were the sites with the lowest number of observed birds. The values of the statistics and *p*-values obtained from pairwise comparisons using post hoc tests are reported in Appendix A.

The most frequently encountered birds belong to synanthropic and generalist species (e.g., Rock Pigeon, Common Wood Pigeon, Italian Sparrow, Starling, etc.) (Appendix A). Sixty-eight species belong to the Passeriformes Order and represented the 52.10% (corresponding to n = 27,539 individuals) of the total wild birds encountered. Columbiformes, with only four species, represent the 27% (n = 14,263 individuals) of the total observed birds. Within this Order, the two most frequently observed species were the Rock Pigeon (n = 5890 individuals) and the Common Wood Pigeon (n = 5726 individuals).

After these, the most represented orders were Charadriiformes (n = 3185) and Anseriformes (n = 2529), representing respectively the 6.37% and 4.79% of the wild birds identified in the ornithological transects. Among the Charadriiformes, the Black-headed Gull (representing 47.06% of the observed Charadriiformes) was the most represented species, although its presence in close proximity to the farms (detected through camera traps) was extremely rare. In fact, this species was photographed only in two farms over a few days during the spring months (n = 9 in May, and n = 5 in June). A total of 17 other Charadriiformes species were recorded. The Mallard was the most abundant within the Anseriformes Order, comprising 81.69% of observed species, while the remaining were uncommon or very rare. Notably, the camera-trapping activities did not detect the presence of any duck species, except for the Mallard.

#### 3.1.1. Correspondence Analysis

The 25 most common species (whose abundance was higher than the 1% of the total birds counted) were considered for the CA to visually assess their relationships with the farming sites (Figure 3a) and sampling months (Figure 3b). The two components explain 54.38% of the total variance observed in the bird species distribution with respect to the farming sites, and 70.65% of the total variance observed with regards to the month. In both cases, a significant association was observed between the sites and species (χ^2^ = 33892, df = 216, *p*-value < 0.0001), and between months and species (χ^2^ = 28805, df = 264, *p*-value < 0.0001). The points’ positions in the CA plot should be interpreted considering both their angle and distance from the origin of the axes. For example, sites and species that lie in the same direction from the origin are positively associated, while those lying in opposite directions are negatively associated; moreover, shorter points’ distances from the plot’s origin represent a more even distribution of the species abundances among the sites, while larger distances are associated with unequal abundances [50].

Figure 3a shows that species belonging to the Passeriformes Order tend to have relatively higher abundances associate to inland farms, with no significant deviation from the axes’ origin, which is indicative of an equal distribution of the species’ relative abundances among farms. When considering the inland farms, most Passeriformes species show a moderate preference for the MN-02, VE-01, VR-01, and VR-02 sites. Conversely, bird species belonging to aquatic orders (Charadriiformes, Anseriformes, Suliformes, Gruiformes, and Pelecaniformes) have relatively higher abundances in farms located along the coast. Clear associations with specific farms are evident for certain species (e.g., Mallard towards the FE-01 farm and the Black-headed Gull towards the RO-01 farm), while other species like the Cattle Egret and the Yellow-legged Gull deviate less from the center of the plot. The only species belonging to the Galliformes Order, the Pheasant, shows higher relative abundances associated with the FE-01 farm compared to all of the others. The three species of Columbiformes (i.e., the Rock Pigeon, the Wood Pigeon, and the Eurasian Collared Dove) are located in three different quadrants of the graph, although they exhibit very high relative abundances in all the study sites.

The CA graph obtained for species and months of the year (Figure 3b) indicates a lesser overall deviation of common species from the center of the plot compared to what was observed for farming sites. The Passeriformes Order is distributed throughout the year, with some species mostly associated with the winter months (i.e., Common Caffinch, Medow Pipit, European Greenfinch and European Robin), while others are in close proximity to the axes’ origins (i.e., Eurasian Magpie, Hooded Crow and Common Starling), indicating more evenly distributed abundances. A third group of Passeriformes species is associated with the spring and summer months and represents the typical nesting species of the study area, such as the Barn Swallow, Eurasian Blackcap and Western Yellow Wagtail. The Charadriiformes and Anseriformes Orders, on the other hand, are mostly associated with the late winter period. Finally, the Black-headed Gull is more strongly associated with the months of October, November, and December.

#### 3.1.2. Non-Metric Multidimensional Scaling and Cluster Analysis

Cluster analysis and nMDS identified three distinct groups of farms (Figure 4) based on the different structures of the observed bird community (considering both species and bird abundance):Cluster 1: a group of 4 farms (FE-01, FE-02, RO-01, and VE-02) located in the eastern zone of the study area, near the coast, potentially influenced by the lagoon ecosystem.Cluster 2: a group consisting of 2 farms (BO-01 and MN-01) located over 100 km apart but sharing the exclusive presence of intensive crops and small inhabited centers nearby.Cluster 3: a third group of 4 farms (MN-02, VE-01, VR-01, and VR-02) distributed in the northern zone of the study area, encompassing the inland ecosystems.

The identified clusters are shown on the map in Appendix A.

#### 3.1.3. Alfa Diversity

The distributions of the number of expected species per cluster (Chao1, Figure 5), representing the monthly variation, showed marked differences between clusters (H_2_ = 23.368, *p*-value < 0.0001; pairwise comparisons through post hoc tests revealed that significant differences exist among all three clusters distribution.) The *p*-values are reported in Appendix A. Cluster 1 has the highest number of expected species throughout the study period, while Cluster 2 has the lowest.

The values of the Shannon entropy index measured for each cluster revealed significant differences between clusters (H_2_ = 19.612, *p*-value < 0.0001). The values of statistics and *p*-values obtained from pairwise comparisons through post hoc tests are reported in Appendix A. Significant differences exist between Cluster 2 and Cluster 1, and between Cluster 2 and Cluster 3, while no significant difference is present between Cluster 1 and Cluster 3. Cluster 2 (BO-01 and MN-01 farms) have the lowest diversity values, indicating a lower level of complexity in the observed bird communities in those sites (i.e., lower richness or greater dominance of a few species).

Pielou’s index (Figure 5) provides information about the evenness of the observed communities and indicates how the overall abundance of individuals is evenly distributed among the observed species, regardless of species richness. The trend of this index in the clusters closely mirrored the trend of the Shannon indices, indicating a greater influence of evenness on diversity compared to species richness. Therefore, a cluster with relatively low species richness can exhibit high diversity if the evenness within that population is high (Cluster 3). The values of the Pielou index measured in each cluster during the observation months revealed significant differences between clusters (H_2_ = 18.595, *p*-value < 0.0001). The values of statistics and *p*-values obtained from pairwise comparisons through post hoc tests are reported in Appendix A. As with the Chao1 index, there is a significant difference between all three clusters. 

### 3.2. Camera Trapping

During 2019, the 20 camera traps operated in the time periods are reported in Appendix A. A total of 40 species of wild birds were identified through camera-trapping activities (Table 2). The majority of species identified through camera trapping (n = 39/40, 97.50%) were also observed during transect surveys. Only the Black-winged Stilt was photographed in the proximity of a single farm (RO-01) but not detected through the transect surveys. A total of 8 out of the 40 observed species (20%) were included in the list of target species for AI surveillance provided by EFSA. Only eight species were found in more than five farms (Table 2), and they were mostly synanthropic species (Eurasian Collared Dove, Eurasian Magpie, Rock Pigeon, Common Blackbird, Carrion Crow). The Cattle Egret was also frequently observed near the premises (specifically, on seven farms). In addition to the Eurasian Magpie (Pica pica), which was detected in nine farms, other target species for AI were observed with some frequency, including the Mallard (observed in four farms) and the Common Buzzard (observed in three farms). All the data related to camera-trapping observations are reported in Table 2, including the number of days each species was observed in each farm. The Eurasian Magpie was observed for a total of 180 days at the RO-01 farm, and it was frequently photographed on other farms as well. The Mallard was observed for only a few days in the proximity of four farms (2 days at BO-01, 2 at FE-01, 4 at RO-01, and 14 at VR-02), and similarly, the Black-headed Gull was photographed for only 2 and 12 days in the RO-01 and VE-02 farms, respectively.

The CA applied to the camera-trap data did not highlight any evident relationship between the wild species and the geographic locations of the study sites, the clusters identified based on the ornithological transect surveys, or the farms’ reared species (Appendix A). However, species belonging to the same Order exhibited, to some extent, preferences for some specific sites. For example, some Charadriiformes were predominantly observed at the BO-01 and RO-01 farms, while most Columbiformes were more frequently seen near the MN-02, VR-01, and FE-01 sites, with the exception of the Rock Pigeon, which was strongly linked to RO-01. Additionally, some corvid species (i.e., Magpie, Crow, and Jackdaw) were frequently recorded at the BO-01 and MN-01 sites.

Figure 6 and Figure 7 provide information on the number of days for each month of 2019 when the eight target species for passive surveillance of AI according to EFSA (Figure 6) were photographed, and the seven most present species (in addition to the Magpie, already included among the target species) that were observed in at least five farms throughout the year (Figure 7). The Great White Egret and the Peregrine Falcon were excluded from Figure 6 as they were poorly represented (the former was photographed only twice in February at the VR-02 farm, while the latter was captured only once at the BO-01 farm in March).

From these graphs, we can observe that some species were captured by the camera traps almost daily throughout the year, while others were captured in rare cases and at specific times of the year. The Magpie was frequently photographed in various farms, especially in the ones belonging to Cluster 1 and 2. The Mallard and Black-headed Gull were photographed during the periods of March to June and May to June, respectively. The most frequently visited months for these two species were April and May. The Collared Dove and Blackbird (Figure 7) were the ones captured more consistently, besides the Magpie. The Collared Dove visited the FE-01 farm (Cluster 1) for 142 days, while the Blackbird visited the VR-02 farm (Cluster 3) for 160 days. The Cattle Egret was photographed on at least one farm for 10 months. In May, it was captured on n = 18/31 different days at the MN-01 farm (Cluster 2), and on n = 16/30 days in November at the RO-01 farm (Cluster 1). Finally, the Common Pheasant, although with lower frequencies, was observed throughout the year in five farms belonging to all of the tree clusters.

## 4. Discussion

The present study highlighted the presence of a total of 147 wild bird species in the study area. Such a high species richness is partly unexpected, considering that Italy is known to host a total of about 550 species of wild birds in its whole territory [51], and considering that the study area was relatively small and characterized by the dominant presence of intensive agriculture, densely populated urban centers, and relatively modest environmental diversity.

Despite the landscape homogeneity, it was possible to identify three distinct clusters of selected sites that accounted for different habitats and significantly different levels of wild-bird population diversity. Cluster 1 (VE-02, RO-01, FE-01 and FE-02) consisted of farms located near the Adriatic Sea, suggesting a diversification of the bird communities living in the coastal areas compared to the ones present inland. Although important, this finding is expected, considering that the lagoon area is a preferred or exclusive habitat for various species of water birds [51]. The other two clusters included more inland farms geographically close to each other. Despite their close proximity, the two clusters presented different environmental characteristics: a greater presence of wooded areas, tree-lined rows, and small orchards in Cluster 3 (MN-02, VE-01, VR-01 and VR-02), and cultivated fields and small residential areas nearby farms belonging to Cluster 2 (BO-01 and MN-01). The latter was the one that exhibited the lowest biodiversity levels. Therefore, it can be hypothesized that, even on a small scale (e.g., the farms in the Mantova province, MN-01 and MN-02, are located just over 3 km apart), the composition of the environment can markedly influence the observable species in a given ecosystem, and a landscape exclusively devoted to agriculture can negatively affect the diversity. In support of this finding, a study conducted in Siberia to assess the response of wild bird populations to the abandonment of cultivated rural areas highlighted how landscapes consisting of arable lands with the exclusive presence of low vegetation negatively impacted the bird species diversity [28]. On the other hand, the higher diversity registered in Cluster 3 resulted in being more associated with an increase in the bird communities’ equitability, rather than the observed species richness. Thus, the presence of arboreal vegetation may favor a more equitable distribution of the terrestrial wild birds, suggesting a greater abundance of the uncommon species living in more conserved environments. Species richness and diversity appeared significantly higher in the agricultural landscape combined with the lagoon ecosystem (Cluster 1). This result is supported by an ornithocenosis survey based on the use of video camera monitoring in a free-range farm located in a high-risk AI zone in the Netherlands. The authors identified 16 Orders of wild birds visiting a farm located 2 km from the northern Dutch coast, along the migratory flyways for aquatic birds [52]. The surrounding area of the farm consisted of pastures for cattle and sheep, and a navigable canal near the free-range area of the farm. As observed above, the intermixing of agricultural ecosystems and wetland areas may positively influence the diversity of wild bird populations. On the other hand, other unexplored factors might influence the propensity of certain species to approach the farms, such as environmental, meteorological, or factors related to specific characteristics of poultry farming sites.

The results of the correspondence analysis on the relationship between wild bird species and months highlighted that the most common species exhibited an equitable distribution of relative abundances across months, indicating a lower variability among seasons compared to the variability differences observed among farm sites. Some exceptions are represented by some bird species that exclusively nest in Italy during the spring–summer period, such as the Swallow and the Blackcap, and other species, such as the Black-headed Gull and the Cormorant, which are mainly encountered during the autumn–winter migrations. Both categories of wild birds have a potential role in the epidemiology of AI: wintering birds in Italy mainly represent a risk path for virus introduction, while nesting birds and resident birds pose a risk for viral amplification within the wild populations during the spring and summer seasons [5,53].

The species that were consistently observed via both ornithological transects and camera-trapping surveys are mostly synanthropic and generalist and belong to the Passeriformes and Columbiformes Orders. Charadriiformes and Anseriformes ranked third and fourth, respectively, among the most abundant. However, transect surveys and camera-trap data were discordant, as only a few species belonging to these two Orders were also photographed in the proximity of farm sites, thus indicating that the probability of encountering Charadriiformes and Anseriformes in the surroundings of the premises can be considered low. Supporting this hypothesis, a Dutch study that used GPS tracking to trace the movement patterns of Mallards demonstrated that the daily movements of this species are repetitive and predictable, and strongly linked to the presence of water, as they mostly move from daytime large waterbodies to nighttime smaller wetland areas where they feed [25]. The need to live in the vicinity of water bodies justifies the limited propensity of these wild bird species to frequent intensive poultry farms, where a significant portion of the surface is paved and frequently visited by humans. These observations are extremely relevant in terms of gaining a deeper understanding of the potential dynamics of avian flu in relation to the wild–domestic interface, and the possible routes of virus introduction and spillover into the domestic compartment. In fact, the limited number of camera-trapped Anseriformes and Charadriiformes species events bolsters the idea that, while waterfowl play an important role in the maintenance and amplification of AIVs in natural environments, other wild bird species might act as the actual bridges between the contaminated areas and poultry. From a temporal perspective, the Yellow-legged Gull and the Mallard showed slightly higher abundances during the late winter period at the ornithological transects, while the Black-headed Gull exhibited a strong association with the autumn period. The populations of the first two species are residential in the study area, whereas the Black-headed Gull is mainly a wintering species in Italy and migrates during spring to the breeding sites in Northern Europe and Asia [51]. During a massive HPAI H5N1 epidemic event in 2022–2023, several hundreds of Black-headed Gulls were affected by the virus, resulting in massive die-offs from the HPAI H5N1 infection all over Europe. The same viral subtype circulating among gull populations was identified in many reported domestic outbreaks, indicating potential direct and indirect introductions from gulls to poultry farms [5]. However, this evidence might reflect the situation where multiple transmission events from the wild to the domestic sector occurred due to an exceptional environmental contamination resulting from the massive involvement of seabirds and/or the concurrence of other actors in the transmission path, such as potential bridge species of wild birds.

While it has been demonstrated that Anseriformes and Charadriiformes are capable of releasing high quantities of infectious virus in their feces during certain life stages and seasons [54,55], the role of other species, including many of those observed in the present work, is not yet fully understood. Moreover, scientific literature provides only partial information on the prevalence of AI infection in these species, mostly derived from experimental studies or investigations conducted in limited geographical regions. The data collected through camera trapping allowed the identification of certain wild bird species that frequently come into contact with poultry farms, based on the number of days and the number of farms in which they were photographed. Apparently, camera-trapped wild birds showed no distinct preferences for any geographic locations or domestic-reared species, although some groups of species (i.e., gulls, doves and corvids) were more frequently observed in the proximity of specific sites. It can be hypothesized that other factors rather than the reared species might contribute to the tendency of certain species to approach certain locations that appear very different from each other. These factors might encompass structural and managerial characteristics of farms (such as the quality of maintenance of the facilities, waste management, the handling of carcasses, etc.), and human activities (e.g., frequency of visits, agricultural activities, application of dissuasion techniques, etc.). These conditions not considered in the present study should be taken into account in future research to evaluate their potential impact on the specific propensity of each species to approach and interact with the poultry facilities.

The species that was recorded on the highest number of days was the Eurasian Magpie. This species was also observed at the ornithological transects with high abundances throughout the monitoring period, confirming its widespread distribution in the entire study area. It is also included in the list of target species for passive surveillance of avian influenza compiled by the EFSA, which confirms its competence as a host for AIVs. The propensity of this species to come into frequent contact with farm sites is linked to the fact that the Eurasian Magpie is a scavenger species and is attracted by the presence of carcasses of other birds and eggs. Several studies conducted in Canada, Germany, and the Netherlands document the constant presence of this species in the vicinity of poultry farms. In the Netherlands, in particular, Eurasian Magpies have been frequently observed through photo trapping while searching for eggs in the outdoor pens of a free-range farm [11,26,52]. In another work from South Korea, an experimental infection of nine Eurasian Magpies with an HPAI H5N1 virus was conducted to investigate the AI competence for this species. All the individuals died and actively shed the virus primarily through the oropharynx and, to a lesser extent, through the digestive tract [56]. Moreover, in the same country, two mortality events were registered in wild Eurasian Magpies infected with the HPAI H5N1 clade 2.5 virus during the HPAI epidemics that occurred between December 2003 to March 2004 [57]. As for Italy, four HPAI H5 cases were confirmed in Magpies between 2022 and 2023, and two cases were recorded between 2020 and 2021 in the same species in other European countries [58,59,60,61,62]. The confirmed susceptibility to AIVs, the large number of subjects observed in the study area, and the propensity to come into close contact with farming environments suggest the need to pay particular attention to the potential role that Eurasian Magpies might play in the spread of AI infection. However, the high susceptibility to infection and low levels of viral shedding prompt the need to further investigate the true competence of this species.

Another species belonging to the Passeriformes Order that was frequently photographed in the present study was the Common Blackbird. Although it does not show any particular synanthropic tendency, this species can be commonly observed in the vicinity of poultry farms, especially during the breeding season, as it utilizes farm structures for nesting, as documented in two studies conducted in Germany and the Netherlands [11,52]. Although any specific information related to the viral competence of this species was scarce or absent in the literature, an experimental study conducted in a group of American Robins, a species that belongs to the same Genus of the Common Blackbird, proved a high susceptibility to the infection sustained by different strains of HPAI H5 clade 2.3.4.4 viruses, also showing high levels of viral shedding [63]. However, only two cases were notified in Blackbirds during 2020–2023 in Europe, one of which was recorded in Italy in 2022 [58,59,60,61,62]. This finding suggests the need to conduct specific studies to further investigate the susceptibility level of the European Blackbird, to better assess its role in the wild–domestic interface, and to determine its actual competence in becoming infected and spreading the virus.

Among the Columbiformes, the Eurasian Collared Dove and the Wood Pigeon were photographed on almost all the camera trapping stations. Interestingly, the Rock Pigeon, the most abundant species according to the ornithological transects surveys, was photographed in four out of ten farms only. These species are of interest due to their potential role as bridge hosts, specifically functioning as mechanical carriers of AIVs [11]. The review by Abolnik, 2014 [64], specifically investigating the risk AIVs introduction by these species highlighted that they are dead-end hosts of infection, capable of viral shedding for a very short period in concentrations below the levels considered necessary for the infection propagation. In 2021–2023, three samples of Collared Dove tested positive in Italy, although no other cases were recorded during the same period in other European countries. As for the Wood Pigeon, five cases were notified in Europe in 2018–2023, two of which were detected in Italy [58,59,60,61,62]. It emerges that their role and impact on the transmission dynamics need to be further investigated, especially considering their massive abundance in the territory and their tendency to come into close contact with farming structures.

The Cattle Egret is the heron species that most easily exhibits synanthropic behaviors [11]. The finding is confirmed in the present study by the camera-trapping survey that frequently registered the presence of the Cattle Egret in the vicinity of the farms. Despite the existence of migratory populations, the majority of the individuals of this species are resident in the Italian territory [51]. This is further supported by the ornithological transects, which did not show any significant associations between the identified individuals and any specific seasons of the year. Several studies have hypothesized that Cattle Egrets may play as a bridging role species for AI, considering their strong presence in wetlands and human-altered areas. In Italy, Cattle Egrets were found positive for AIVs on two occasions in 2022 and 2023. Furthermore, several wild birds belonging to the Pelecaniformes Order tested positive in Europe: 21 *Ardea cinerea* and 14 *Ardea alba* in 2021, and 2 *Ardea cinerea* in 2020 [58,59,60,61,62]. Moreover, in Egypt, an HPAI H5 virus was isolated from a Cattle Egret in a sample of sixty individuals collected near a poultry farm resulted infected with the same virus [65]. In Vietnam, six Cattle Egrets were experimentally infected with an HPAI H5N1 virus via intranasal inoculation to investigate their possible epidemiological role in transmitting the infection to domestic poultry [66]. The Cattle Egrets proved to be highly susceptible to the infection and died within a week of the experimental inoculation. On the contrary, five chickens that were in contact with the infected Cattle Egrets eight days after infection were found healthy, with no signs of illness nor mortality. According to the authors, this may be due to the low levels of the viral shedding sustained by the Cattle Egrets. These findings suggest a limited involvement of this species in the spread of the infection within the domestic sector. However, further studies involving different experimental conditions and focusing on poultry species known to be more susceptible (such as turkeys) would be useful to definitively exclude the role of this species in the disease epidemiology.

Another species of interest, considering its strong tendency to approach the domestic poultry farms, is the Pheasant. Pheasants could represent a potential risk for the transmission of AIVs, as farms can be highly attractive to wild birds in search of food resources. It should be noted that the presence of Pheasants in close proximity to premises might be related to the practice of releasing new individuals into the wild environment for hunting purposes and gaming activities, usually during the winter season. These pen-raised animals for restocking purposes are accustomed to the presence of humans and their activities, thus resulting in an unnatural tendency to remain in close proximity to the places where they were raised in the early part of their lives. Pheasants can also be efficiently infected with AIVs, as highlighted by the high number of cases reported in recent years in Europe (161 between 2018 and 2021), as well as 11 HPAI outbreaks in reared Pheasants in Italy between 2021 and 2023 [58,59,60,61,62]. A serological survey conducted in Italy between 1995 and 2002, where a seroprevalence of 12.30% and two seroconversions were registered among 219 sampled wild individuals [67], represents further evidence of its potential role as bridge species. Finally, a Chinese study investigated the pathogenesis and transmission of an HPAI H5N6 virus in pheasants and chickens and revealed an efficient viral transmission between these two species [68].

The camera-trapping methodology applied in this study allowed new insights into the identity of certain species that appear to be particularly inclined to approach poultry farms, allowing for the generation of new hypotheses needed to further investigate their epidemiological role in the transmission of AIVs to the domestic sector. However, the sampling activities carried out via camera traps had some limitations. Firstly, the installed camera traps had a restricted field of view that could not cover the entire area of the farm, which means that the recorded visits only provided partial data of the animals that actually approached the farms. For the same reason, no distinction could be made between the records obtained for the same subjects that moved multiple times in front of the range of the camera traps, thus leading to a potential overestimation. Another important consideration concerns the aspect of the wild bird species and their behavior. A bird with distinctive and large-sized plumage or more accustomed to human presence is usually more easily observable and recognizable compared to a smaller bird with a non-distinctive plumage or timid temperament. Moreover, in the case of uncertain or impossible identification of the recorded birds, data were excluded from subsequent analyses, which may have led to an underestimation of certain species. The study of the potential bridge species should account for not only the species abundances and visiting occurrences, but also their duration, as pointed out in the study of Elbers et al., 2020 [52]. The focus of the Dutch study was the identification and quantification of the interactions occurring between wild and domestic birds through the analysis of video camera monitoring data installed in a single free-range farm. As regards the present study, such an approach would have been hardly applicable considering the higher number of sites and premises involved. For this reason, it was not possible to derive reliable quantitative data from the camera-trapping activities. Regarding the counting of wild birds via ornithological transects, a significant risk, also remarked by Burns et al. (2012) [26], is the potential underestimation of well-camouflaged species, birds that do not vocalize, and species that are active only during specific times of the day. As an example, the Little Owl was detected much more frequently via camera traps compared to the extremely rare sightings experienced by the ornithologists, considering that the on-field surveys were conducted exclusively during daylight hours.

The findings that emerged in the present work emphasize the need to further investigate the virological competence of the wild bird species that have shown the greatest affinity with the farm environments. A deep analysis of the species’ virological competence would consider the quantification of the incubation period, the viral shedding, and the viral titers eliminated through both the respiratory routes and the digestive system. Concurrently, an assessment of the dynamics of the interface occurring between the maintenance and amplification hosts and the potential bridge species that share the same natural environments would provide crucial information to tackle the more elusive aspects of AI ecology. It would also be relevant to disentangle the complex epidemiology behind the circulation of AIVs in the wild populations, which can involve new introductions during the autumn and winter periods and/or the spread among nesting and resident species during the spring and summer seasons. Lastly, the development of statistical and mathematical models would be meaningful to analyze how different bird community structures might qualitatively and quantitatively influence the probability of spillover events into the domestic sector.

## 5. Conclusions

This study aimed to characterize the bird communities in the surroundings of poultry farms in northeastern Italy that could contribute to a better understanding of the ecology and biology of certain bird species serving as bridge hosts between wild waterfowl, reservoirs of influenza viruses, and domestic poultry. To achieve this goal, data on the presence and abundance of bird species in the vicinity of ten poultry farms were collected and analyzed to characterize the biodiversity levels and potential spatio-temporal differences. In the future, the preliminary findings of the present study can be integrated with epidemiological data to evaluate the bird species that might play a bridging role in the ecosystem, as well as their impact on the probability of emergence of new outbreaks in poultry, while also taking into account ecological factors, such as social behavior, their tendency to mix with other species, the proportion of young individuals, and feeding habits [27,69].

The main results obtained relate to the species richness, which was surprisingly higher than expected in the considered study area, and a significant diversification of the bird communities in different sites and habitats. The most interesting observations concerned the high frequency of farm visits by the Eurasian Magpie, the Common Blackbird, the Cattle Egret, the Common Pheasant, and the main species of pigeons (i.e., the Eurasian Collared Dove and the Wood Pigeon). Furthermore, the concept related to the low propensity of Charadriiformes and Anseriformes species to come into close contact with poultry farms was confirmed, especially during the autumn and winter seasons.

Overall, this study provides: (i) an in-depth characterization of the bird communities in northeastern Italy and factors that may significantly influence their composition; (ii) the remark that Anseriformes and Charadriiformes species show limited direct contact with intensive poultry farming sites, and (iii) the identification of the bird species that are particularly inclined to come into proximity with farming environments. Moreover, this investigation lays the foundation for directing future studies toward specific wild bird species or geographical areas considered to be at higher risk of virus spread and transmission.

## Figures and Tables

**Figure 1 microorganisms-11-02601-f001:**
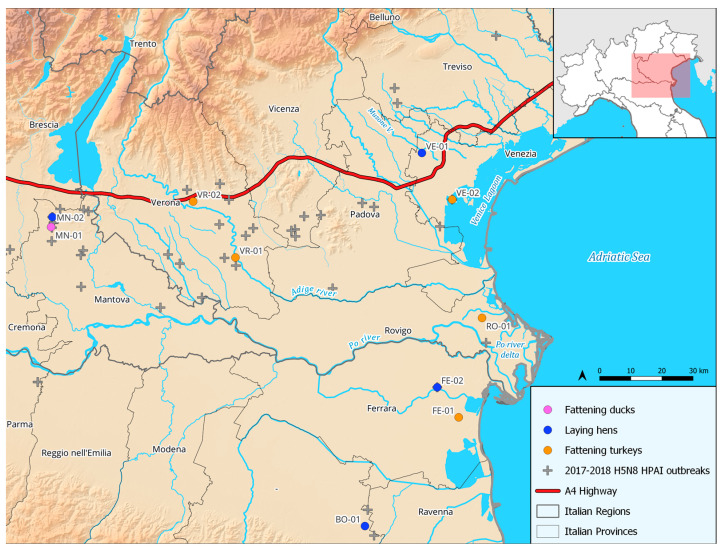
Farms’ locations included in the study. In grey: domestic and wild outbreaks reported in the study area during the HPAI H5N8 epidemic of 2017–2018.

**Figure 2 microorganisms-11-02601-f002:**
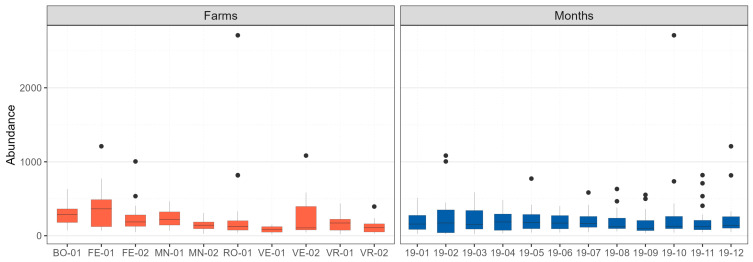
Overall abundances of the observed wild birds are presented. (**Left**) points to abundance distributions per farming site; (**right**) points to abundance distributions per month. The middle, lower and upper hinges of the box plots represent the 50%, 25% and 75% quantiles, respectively; the black dots represent the outliers.

**Figure 3 microorganisms-11-02601-f003:**
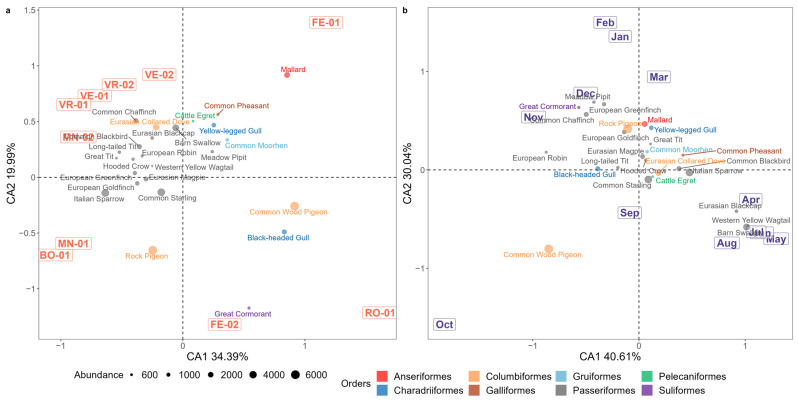
Correspondence analysis plots are presented: (**a**) association between sites and common species; (**b**) association between months and common species. The species are colored according to the Order to which they belong, and the size represents their absolute abundance. CA plot points’ positions reveal associations based on their angle and distance from the axes’ origin: same direction for positive, opposite for negative associations; shorter distances for even species distribution, longer for unequal abundances.

**Figure 4 microorganisms-11-02601-f004:**
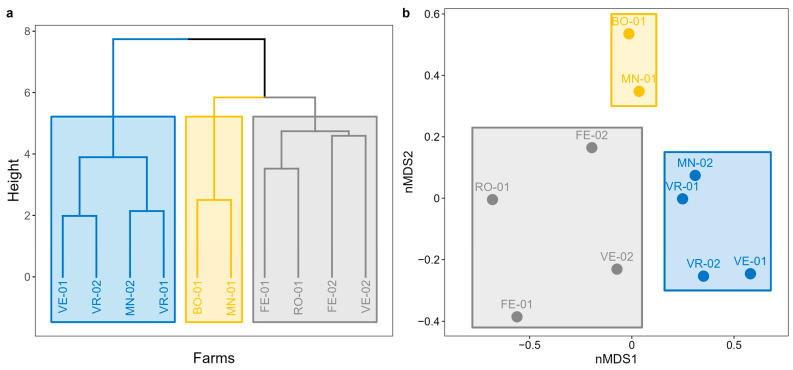
(**a**) Dendrogram and (**b**) nMDS according to the Bray–Curtis distance (stress: 0.09) are presented. Grey represents Cluster 1, yellow, Cluster 2, and blue, Cluster 3.

**Figure 5 microorganisms-11-02601-f005:**
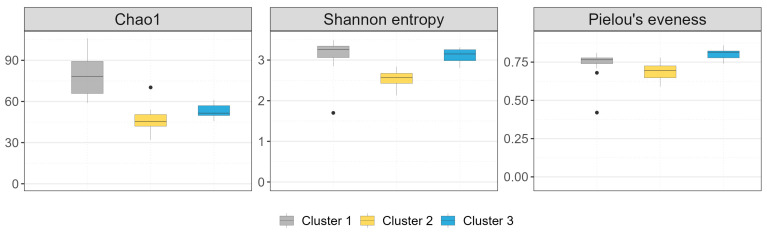
Diversity indices (Chao1, Shannon entropy and Pielou’s eveness) distributions for the three identified clusters are presented. The middle, lower and upper hinges of the box plots represent the 50%, 25% and 75% quantiles, respectively; the black dots represent the outliers.

**Figure 6 microorganisms-11-02601-f006:**
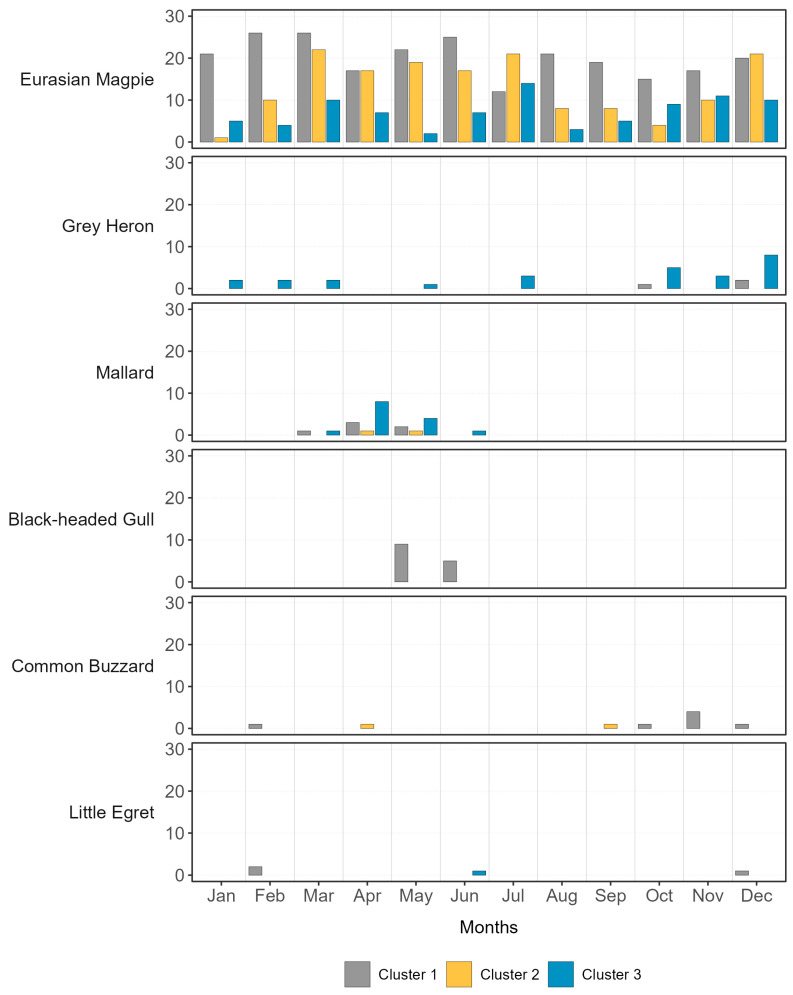
Number of days in which the target species for passive surveillance of avian influenza (EFSA) were observed in more than one farm during the study period in each farm. The Great White Egret and the Peregrine Falcon were excluded from the figure as they were poorly represented. Data were aggregated according to the clusters identified by the nMDS method applied to the transects’ data.

**Figure 7 microorganisms-11-02601-f007:**
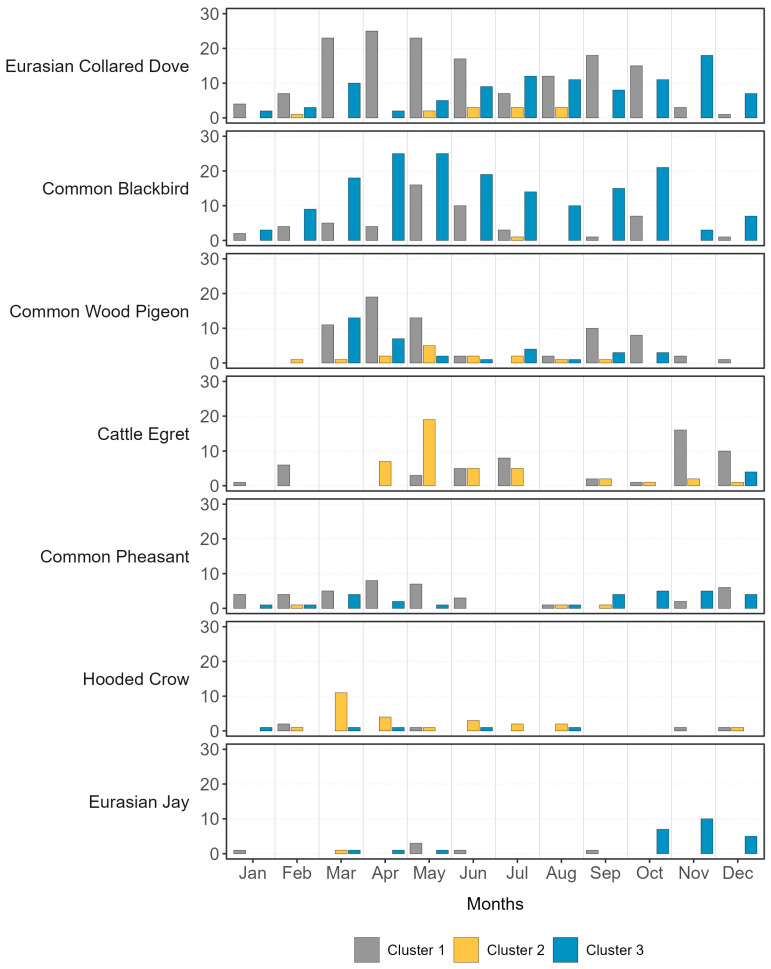
Number of days in which the seven camera-trapped species (including Magpie among the target species) observed in at least five farms were camera-trapped during the study period. Data were aggregated according to the clusters identified by the nMDS method applied to the transects’ data.

**Table 1 microorganisms-11-02601-t001:** The α-diversity indices used to describe the composition of bird communities observed along transects. It includes their description and numerical range.

Index	Description	Numerical Range [min; max]
Species richness (S_obs_)	The number of observed species	[0; +∞]
Chao1	A non-parametric index that estimates the expected number of species in an area. The observed species richness is corrected for the number of undetected species, which is estimated based on the distribution of rare species [38].	[0; +∞]
Shannon entropy	It quantifies both biodiversity and species dominance [39].	[0; ln(S_obs_)]
Pielou’s evenness	It indicates how evenly each species is represented within the population under study [40].	[0; 1]

**Table 2 microorganisms-11-02601-t002:** Wild bird species camera-trapped in the proximity of the 10 farm sites are presented. The observation of each species through camera trapping in each farm is indicated by the number of days the species was photographed in each farm. The last column shows the total number of farms each species was photographed at. Asterisk (*) identifies the camera-trapped wild bird species that belong to the list of 50 target species for passive surveillance established by EFSA.

Common Name (Scientific Name)		BO-01	FE-01	FE-02	MN-01	MN-02	RO-01	VE-01	VE-02	VR-01	VR-02	No. Farms
Eurasian Collared Dove (*Streptopelia decaocto*)		3	142	3	9	26	4	1	15	21	55	10
Common Wood Pigeon (*Columba palumbus*)		2	57	5	13	28		4	6	2	3	9
Eurasian Magpie (*Pica pica*)	*	106	4	3	78	65	180		104	19	8	9
Common Blackbird (*Turdus merula*)			13	38	1	26	2	7	1	5	160	9
Cattle Egret (*Bubulcus ibis*)		17		13	25		38		6		4	6
Hooded Crow (*Corvus cornix*)		19			6	3	5	1		1		6
Common Pheasant (*Phasianus colchicus*)		3	28			26			14		2	5
Eurasian Jay (*Garrulus glandarius*)		1	5	1				3			22	5
Mallard (*Anas platyrhynchos*)	*	2	2				4				14	4
Rock Pigeon (*Columba livia*)		5		7		1	121					4
Common Starling (*Sturnus vulgaris*)		2	24				12		25			4
Little Owl (*Athene noctua*)							4		1	7	21	4
Common Chaffinch (*Fringilla coelebs*)		1				1					2	3
Yellow-legged Gull (*Larus michahellis*)		14					9		2			3
Eurasian Tree Sparrow (*Passer montanus*)		4	10				1					3
Common Buzzard (*Buteo buteo*)	*	2	6						1			3
Grey Heron (*Ardea cinerea*)	*		2						1		26	3
Little Egret (*Egretta garzetta*)	*		2						2		1	3
European Robin (*Erithacus rubecula*)			3	1		12						3
European Turtle Dove (*Streptopelia turtur*)			1		1	7						3
European Green Woodpecker (*Picus viridis*)					1	2					7	3
Common Kestrel (*Falco tinnunculus*)									1	2	1	3
Barn Swallow (*Hirundo rustica*)		3						1				2
Common Redstart (*Phoenicurus phoenicurus*)			2								1	2
Common Chiffchaff (*Phylloscopus collybita*)			1			1						2
Italian Sparrow (*Passer italiae*)			15			1						2
European Goldfinch (*Carduelis carduelis*)						1	1					2
White Wagtail (*Motacilla alba*)							9				9	2
Black-headed Gull (*Chroicocephalus ridibundus*)	*						12		2			2
Peregrine Falcon (*Falco peregrinus*)	*	1										1
Common Stonechat (*Saxicola torquatus*)		1										1
Eurasian Jackdaw (*Corvus monedula*)		5										1
Eurasian Hoopoe (*Upupa epops*)			10									1
Eurasian Blue Tit (*Cyanistes caeruleus*)						1						1
Black-winged Stilt (*Himantopus himantopus*)							7					1
Mediterranean Gull (*Ichthyaetus melanocephalus*)							6					1
Great Egret (*Ardea alba*)	*								2			1
Common Swift (*Apus apus*)									1			1
Purple Heron (*Ardea purpurea*)											1	1
Common Moorhen (*Gallinula chloropus*)											2	1
Richness		18	18	8	8	15	16	6	16	7	18	

## Data Availability

The data presented in this study are available upon request from the corresponding author.

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
