# Peer review of "Study of the Interface between Wild Bird Populations and Poultry and Their Potential Role in the Spread of Avian Influenza"

_microorganisms, 2023, doi:10.3390/microorganisms11102601_

Round 1

Reviewer 1 Report

Martelli et al, studied the interface between wild bird populations and poultry and their potential role in the spread of Avian Influenza. The results highlighted the importance of ongoing research on the wild-domestic bird interface, advocating for a wider range of species to be considered in AIVs surveillance and pre- 26 vention programs. The study is very interesting, and well-written. I recommend its publication.

Only a minor comment, please revise the use of abbreviations throughout the manuscript, examples: line 33: AI viruses  ----should be AIVs, Line 133: highly pathogenic avian influenza-------Should be HPAI. 

Minor editing of English language required

Reviewer 2 Report

Comments to authors

    The authors investigated what kinds of bird species approached to poultry farms in Northeastern Italy for the purpose of estimating their potential role in disseminating high pathogenicity avian influenza viruses. The context is very clear, and the manuscript is well organized. The manuscript is expected to provide valuable information for countries and regions suffering from avian flu outbreaks. I would like to suggest several points as follows to make this article more significant and to make readers easily understood.

Major comments

1. The relationships between the observed bird species and the poultry species (ducks, hens, and turkeys) are not sufficiently discussed. Regardless of the presence or absence of their correlations, the related discussion should provide valuable information to readers and make the paper more significant.

2. ‘MN-01’ and ‘MN-02’ showed different results whereas these two farms are closely situated, and these farms were classified into different Clusters 2 and 3, respectively. The authors should show their interpretation of the results.

Minor comments

L46, L59, L69

Highly pathogenic avian influenza was renamed “High pathogenicity avian influenza” by WOAH (OIE). The term ‘high’ should be used instead of ‘highly’. 

Highly pathogenicity” may be acceptable, but I recommend to revise to improve your article.

OIE Terrestrial Animal Health Code and Manual have been already updated as follows.

<OIE Terrestrial Animal Health Code> (https://www.oie.int/fileadmin/Home/eng/Health_standards/tahc/current/chapitre_avian_influenza_viruses.pdf)

<OIE Terrestrial Manual> (https://www.oie.int/fileadmin/Home/eng/Health_standards/tahm/3.03.04_AI.pdf)

L69, L123

The abbreviation ‘HPAI’ should be used because the authors already designated it in L46.

L89

Please designate the reference. ([REF])

Figure 1

Please add the names of places which the authors used in the manuscript, especially in ‘2.1 Definition of the study area’. For example, rivers, coast, lagoon names etc. These should be helpful for readers to understand the circumstances in the study area.

L207

Table S1 Table S2?

Figure 2

Please add explanation about the black dots in the figure to its legend.

Figure 2 legend

per month per farming site

(left and right are reversed)

L286

…Starling, etc.) (Table S2)

(‘(Table S2)’ should be added.)

L326

Black-backed Gull Black-headed gull

L377-L378

‘Cluster 2 and the other two clusters,’ ‘Cluster 1 and the other two clusters,’ ??

‘Cluster 1 and Cluster 3’ ‘Cluster 2 and Cluster 3’ ??

L409

‘including the Mallard, observed in 4 farms)’ ‘including the Mallard (observed in 4 farms)’

Figure 5

Please add explanation about the black dots in the figure to its legend.

Table 3

Please align the lines correctly. For example, ‘European Green Woodpecker’, ‘Black-headed Gull’, ‘Mediterranean Gull’ etc.

(It might be the matter of the editorial office)

Reviewer 3 Report

The manuscript investigates the role of water birds in spreading Avian Influenza viruses (AIVs) and explores the possibility that other wild birds act as bridging hosts, introducing AIVs to poultry. The study employs ornithocoenosis analysis, census transects, and camera-trapping to characterize bird communities. Through multivariate analyses, the study reveals a complex ornithic population with significant variations in wild bird species composition. One highlighted finding is identifying certain bird species in close proximity to poultry premises, suggesting their potential role in transmitting AIVs to poultry. This work highlights the importance of ongoing research on the wild-domestic bird interface and target bird species in AIV surveillance and prevention.

One main comment is whether the authors have confirmed AIV infections (sequencing or serological tests) in those potentially important bridging species, which could provide solid evidence to support their conclusions.

Some minor changes should be addressed in the text:

1.     Line 89 missing reference.

2.     Line 299-301 rephrase this sentence.

3.     Line 312-317 should be put in the Figure legend.

Overall English quantity is high
